# Familial CCM Genes Might Not Be Main Drivers for Pathogenesis of Sporadic CCMs-Genetic Similarity between Cancers and Vascular Malformations

**DOI:** 10.3390/jpm13040673

**Published:** 2023-04-17

**Authors:** Jun Zhang, Jacob Croft, Alexander Le

**Affiliations:** Departments of Molecular & Translational Medicine (MTM), Texas Tech University Health Science Center El Paso (TTUHSCEP), El Paso, TX 79905, USA

**Keywords:** Cerebral cavernous malformations (CCMs), CCM signaling complex (CSC), familial CCM (fCCM), sporadic CCM (sCCM), phosphatidylinositol-4, 5-bisphosphate 3-kinase catalytic subunit p110α (PIK3CA), tumor driver mutations, tumor passenger mutations, gain-of-function (GOF), PIK3CA-related overgrowth spectrum (PROS), vascular malformations (VMs), venous malformations (VeMs), developmental venous anomalies (DVAs)

## Abstract

Cerebral cavernous malformations (CCMs) are abnormally dilated intracranial capillaries that form cerebrovascular lesions with a high risk of hemorrhagic stroke. Recently, several somatic “activating” gain-of-function (GOF) point mutations in PIK3CA (phosphatidylinositol-4, 5-bisphosphate 3-kinase catalytic subunit p110α) were discovered as a dominant mutation in the lesions of sporadic forms of cerebral cavernous malformation (sCCM), raising the possibility that CCMs, like other types of vascular malformations, fall in the PIK3CA-related overgrowth spectrum (PROS). However, this possibility has been challenged with different interpretations. In this review, we will continue our efforts to expound the phenomenon of the coexistence of gain-of-function (GOF) point mutations in the *PIK3CA* gene and loss-of-function (LOF) mutations in *CCM* genes in the CCM lesions of sCCM and try to delineate the relationship between mutagenic events with CCM lesions in a temporospatial manner. Since GOF PIK3CA point mutations have been well studied in reproductive cancers, especially breast cancer as a driver oncogene, we will perform a comparative meta-analysis for GOF PIK3CA point mutations in an attempt to demonstrate the genetic similarities shared by both cancers and vascular anomalies.

## 1. Introduction

Cerebral cavernous malformations (CCMs) and developmental venous anomalies (DVAs) are two types of cerebral vascular malformations characterized by abnormal blood vessels in the brain. CCMs are clusters of abnormal capillaries, while DVAs are irregular arrangements of cerebral veins. While most familial cases of CCM (fCCM) are caused by gene mutations, the cause of most sporadic cases (sCCM) remains unknown. However, activating mutations in the gene PIK3CA have been recently linked to the pathogenesis of CCMs [1,2,3,4]. There is ongoing debate about the relationship between CCMs and DVAs and their potential interaction [5]. This review aims to examine the relationship between altered PI3K signaling pathways and the formation and progression of sporadic CCMs, drawing on data from tumorigenic genetics in reproductive cancer, specifically breast cancer.

The transforming potential of PI3K catalytic isoforms was realized decades ago in cancer research [6,7,8], and these isoforms have been defined as oncogenes [8,9]. Somatic GOF PIK3CA point mutations have been frequently described in various cancers as an oncogene [10,11,12,13,14,15], as well as in many forms of vascular anomalies, especially the well-known vascular malformations (VMs) [16,17,18,19,20]. This shared genetic similarity between tumors and vascular anomalies suggests that as benign tumors, VeMs might have similar genetic underpins as cancers caused by the exact same somatic PIK3CA GOF mutations [16,19,20,21]. 

CCMs (or cavernous angiomas) are clusters of tightly packed, abnormal cerebral capillaries with thin walls, while DVAs (or venous angiomas) are irregular arrangements of cerebral veins, and DVAs are the most common form of congenital cerebral venous malformations (VeMs) without any clinically symptomatic event [22]. It has been well defined that most fCCM cases are caused by mutations in one of the three *CCM* genes, while the majority of sCCM cases are still unknown, despite that *CCM* gene mutations have been found in some sCCM [23,24]. A common clinical phenomenon of sCCM is its frequent association with DVA [25,26,27]. Recently, activating PIK3CA ‘hot spot’ point mutations (DVAs/VeMs causative mutations) has been proposed as the possible fourth locus for the pathogenesis of CCMs [4]. However, the activation of PI3K pathways has not been found to be an universal regulator of vascular morphogenesis, especially for CCMs. Furthermore, following reports on the coexistence of CCMs and DVAs, more attention has been given to the potential interaction and relationship between CCMs and DVAs. One vessel structural/fluid dynamic hypothesis suggested that either CCMs or DVAs can directly or indirectly affect their adjacent area venous outflow through their malformed cerebral vessels, which could further exacerbate pre-existing structural DVAs followed by further worsened outflow. Under prolonged exposure to altered venous outflow, DVAs can eventually induce hemorrhagic events in adjacent CCM lesions [28]. Another genetic hypothesis suggested that ECs of DVAs with GOF mutations in the *PIK3CA* gene have high somatic mutation potential as the known tumor driver mutation. These clonal ECs of DVAs with GOF PIK3CA mutations are hyper-proliferative while migrating along vasculatures, leading to the occurrence of secondary mutations, including *CCM* genes, in these ECs, which then result in the formation and progression of sporadic CCM lesions at DVAs adjacent vessels [5].

This review aims to examine the relationship between altered PI3K signaling pathways and the formation and progression of sporadic CCMs, drawing on data from tumorigenic genetics in reproductive cancer, specifically breast cancer.

## 2. Common Somatic Activating GOF Mutation of *PIK3CA* Gene Is the Key for Tumorigenesis

*Spontaneous somatic mutation is a common event.* Genetic mutations in somatic cells are generally random and can contribute to aging, neurodegeneration, and cancer initiation [29,30]. These mutations occur in every individual tissue and accumulate over time, leading to clonal expansion of cells with driver mutations in either cancer [31,32,33] or non-cancer tissues [34,35]. Although most spontaneous mutations have no noticeable effect, some mutations can alter key cellular functions, and, with time, the accumulation of these key mutations can lead to the growth advantage of cells, which is the first step in tumorigenesis [36,37,38,39]. Hereditary oncogenic genes, which are rare germline mutations in DNA repair pathways, increase the risk of cancer [40,41], but most cancers are caused by somatic mutations in oncogenic genes [30,42,43,44,45,46,47]. Studies have shown that certain genetic mutations confer a selective growth advantage [48,49] and can be considered cancer drivers [31,33], suggesting that most cancers arise from “bad luck” random events [48,50,51,52,53,54]. Cancer driver mutations are responsible for oncogenesis, and many tumors have recurrent somatic mutations in the PI3K-AKT1-mTOR signaling pathways [9,55,56,57], making these PI3K signaling genes good candidates for cancer therapy [57,58,59,60,61,62,63,64,65,66,67,68,69,70].

*PI3K/AKT/mTOR signaling pathway and cancer.* The spontaneous somatic mutations of specific oncogenic genes, including the PI3K/AKT/mTOR signaling pathway, are crucial in regulating cellular processes such as survival, growth, differentiation, metabolism, and cytoskeletal reorganization [59,60,61]. These mutations are a major cause of cancer [59,71]. As an evolutionary conserved family of lipid kinases [71,72], the PI3K pathway is one of the most frequently activated in human cancer [30,42,43,44,45,46,47], where it phosphorylates phosphatidylinositides to regulate downstream signaling [73,74]. One of the most significant transforming potentials from deregulation of class I PI3K is reflected in the *PIK3CA* gene [12,14,15,56,75,76,77,78,79,80,81,82,83]. 

*Somatic activating* *GOF* *PIK3CA **point mutation is a tumor driver*. Genomic amplification of both *PIK3CA* and *AKT* genes has been linked to early stages of tumor formation [12,80,84,85,86,87,88,89]. Activating mutations in these genes have been frequently identified in many types of cancer and have been shown to result in hyper-activation of the PI3K/AKT/mTOR signaling pathway [14,15,56,76,77,78,79,80,81,82,83]. This leads to a range of altered cellular processes, including survival, proliferation, growth, metabolism, angiogenesis, and metastasis [83,90,91,92]. GOF point mutations in the *PIK3CA* gene have been identified as causative oncogenic mutations in various cancer types, particularly in reproductive cancers [14,57,91,93,94,95,96], making *PIK3CA* one of the major cancer driver genes [14,15,56,76,77,78,79,80,81,82,83]. Of the three class-I PI3K catalytic isoforms, only PIK3CA has been shown to be essential for vascular development, and its expression is selective in endothelial PI3K signaling during angiogenesis [90,97,98,99]. This has been supported by numerous clinical reports that show the same GOF point mutations in *PIK3CA* gene are present in both tumors and vascular anomalies [3,4,98,99,100,101,102,103,104,105,106,107].

## 3. *Somatic GOF PIK3CA* Mutations Are Oncogenic and Driver in Breast Cancer

*Oncogenic mutations in the PI3K signaling pathway and their role in breast cancer.* DNA sequencing of cancer samples revealed that among the 16 members of the PI3K family, the *PIK3CA* gene is the only one with activating (GOF) mutations [12,75,108,109,110], making it oncogenic [12,75]. This was suggested by the high frequency of GOF *PIK3CA* mutations in breast cancer cases [111,112]. Additionally, activating mutations in other genes in the PI3K signaling pathway, such as AKT1, are also commonly found in breast cancer [112]. Oncogenic mutations in the PI3K kinase signaling pathway generally refer to activating mutations in key members such as PIK3CA and AKT1, as well as other well-known oncogenic genes like *TP53* [91,113,114,115]. The hyper-proliferative nature of these mutations leads to genomic instability and an increased number of secondary mutations in the cancer cells during the evolution of the cancer. Some of these passenger mutations might contribute to tumor progression [116,117,118], while others are not found to have a significant impact or their genetic role is not yet fully understood [54,116,119,120,121].

PIK3CA * and TP53 oncogenes as key drivers of breast cancer tumorigenesis.* Both *TP53* and *PIK3CA* somatic mutations have been recorded as the top-two recurrent oncogenes in breast cancers [111,122,123,124,125,126], with *TP53* dominant in mutations (69%), followed by *PIK3CA* somatic GOF mutations (29%), indicating their oncogenic driver roles in breast cancer tumorigenesis [127,128].

The oncogenic effect of enhanced PI3K signaling in the reproductive system, especially in breast cancers, has been extensively studied [91,129,130,131]. Cooperation between somatic GOF *PIK3CA* and LOF *p53* mutations have been well documented in breast cancer [19,125,132,133,134,135,136,137] and reproductive cancers in general [19,82,125,133,134,135,136,137,138,139,140,141,142]. 

*GOF* *PIK3CA mutations induce the passenger mutations of members of CmPn signaling in reproductive cancers*. The treatment of PIK3CA-related breast cancers has been a major focus in breast cancer research, and the combination of PI3K/AKT inhibitors with rapamycin has been explored as a potential treatment option [63,114,143,144,145,146]. Additionally, this approach has also been applied to the treatment of patients with vascular anomalies and overgrowth syndromes caused by GOF PIK3CA mutations, indicating a genetic similarity between these two conditions [16,147,148,149].

It has been defined that the three CCM proteins form the CCM signaling complex (CSC) that couples classic and non-classic progesterone (PRG) receptors (nPRs/mPRs) to form the CSC-mPR-PRG-nPR (CmPn) signaling network, which has a major impact on angiogenesis and tumorigenesis [150,151,152,153,154,155,156,157]. Key members of the CmPn network have been defined as important biomarkers for certain oncogenic conditions [150,151,152,155,156,158,159,160,161,162]. Since most common GOF *PIK3CA* mutations (hotspots) in breast cancers are somatic point mutations, by utilizing the GDC data portal from the National Cancer Institute (NCI), simple somatic mutations (SSM) of breast cancer lesions for *TP53* and *PIK3CA* genes, along with all key CmPn signaling components, were assessed with Oncogrid. The distribution of SSM for breast cancer lesions clearly demonstrates that both *TP53* and *PIK3CA* genes are indeed breast cancer drivers, while all SSM for *CCM* genes and other members of the CmPn network are passenger mutations (Figure 1). This finding was further validated by the similar distribution of SSM of reproductive cancer lesions for *TP53* and *PIK3CA* genes, and key CmPn signaling components with Oncogrid (Figure 2), suggesting the oncogenic role of GOF *PIK3CA* mutations (hotspots) to generate randomly appearing passenger gene mutations of key CmPn signaling components. Therefore, as previously indicated, the secondary passenger mutations of *CCM* genes may have consequential events in sporadic CCMs [5].

Confirmation of oncogenic driver role of GOF PIK3CA mutations in mouse cancer models and their association with breast cancer subtypes and vascular phenotypes. Combining with other oncogenes, activating PIK3CA mutations in transgenic mice were utilized to create various types of cancer mouse models (such as colon, prostate, lung, and brain cancers, etc.) [163,164,165,166,167,168,169,170], in either a non-inducible [96,125,171] or conditional and inducible system [79,95,132,163,167]. Activating GOF PIK3CA mutations have been shown to predominantly predispose mouse reproductive organs/tissues to tumors [77,79,95,96,124,125,132,171,172,173,174]. Actually, the majority of mouse models with activating GOF PIK3CA mutations have been used to investigate various subtypes of breast cancer and their therapeutic strategies [77,78,79,96,124,125,132,164,171,172,173,175,176,177,178]. Interestingly, although like other tumor mouse models, breast cancer mouse models with enhanced PI3K activity targeted in epithelial cells, still exhibit vascular phenotypes such that both lymphatic (LMs) and venous malformations (VeMs) and have been reported in epithelial cancers associated with GOF PIK3CA mutations [72,101], suggesting a shared genetic mechanism between breast cancers and vascular anomalies. 

## 4. *Somatic GOF PIK3CA* Mutations Are Genetic Drivers for Vascular Anomalies

*Vascular anomalies are common phenotypes in the PIK3CA-Related Overgrowth Spectrum (PROS).* PIK3CA mutations are widely recognized as a common human oncogene and have been detected in many types of cancer [9,12,60]. GOF point mutations in the *PIK3CA* gene, such as E542K, E545K, and H1047R, result in the activation of the PI3K pathway and are considered to be oncogenic [76]. However, PIK3CA mutations also cause noncancerous conditions known as PIK3CA-Related Overgrowth Spectrum (PROS) disorders, which are characterized by hyper-proliferation in various tissues and cell types [19,179,180]. It is noteworthy that vascular anomalies are a prevalent phenotype in PROS [98,101,181,182], emphasizing the critical involvement of PIK3CA mutations in their development [98,183,184,185]. The presence of these mutations has been confirmed in various forms of vascular anomalies [98,183,184,185], indicating their contribution to the maintenance of microvasculature. 

*Somatic GOF PIK3CA mutations is a driver of various types of vascular malformations (VMs).* Vascular malformations (VMs) are a group of defects in the formation of blood vessels and are classified into lymphatic, capillary, venous, and arteriovenous malformations [186,187,188,189]. These can further be divided into simple, complex, and syndromic VMs based on their vessel composition [188,190,191]. *PIK3CA*, a gene affecting cellular growth, is crucial for proper vascular development [97]. Mutations in PIK3CA result in hyper-proliferative blood vessels with reduced pericyte coverage and decreased expression of arteriovenous markers, leading to the development of PIK3CA-Related Overgrowth Spectrum (PROS) disorders [180,182,192,193,194], with VMs being a common phenotype [195,196,197]. Hyper-proliferative clonal populations with enhanced PI3K/AKT signaling have been isolated from patients with simple and complex VMs and are caused by mutations in PIK3CA [16,20,101,102,198,199,200,201]. Complex and syndromic VMs often result from mosaicism in early developmental stages and can vary in manifestation depending on the timing of the mutation and the cell lineage affected [190,202]. Despite these differences, all GOF PIK3CA-associated VMs share the same pathogenesis as slow-flow VMs (SFVMs), which are frequently caused by GOF PIK3CA mutations [18,190,203]. This has been supported by recent sequencing data in patients with slow-flow VMs [18].

*VeM pathology can be realized in mouse models carrying GOF PIK3CA mutations.* Creating mouse models with activated PIK3CA mutations for vascular phenotypes has been challenging. Early attempts using the constitutively active form of membrane-bound PIK3CA resulted in embryonic lethality, indicating that too much activation of PI3K signaling can cause lethal vascular malformation (VM) phenotypes [96]. This embryonic lethality occurred due to defective vasculogenesis and angiogenesis when activated early in embryogenesis, suggesting that over-activation of PI3K signaling too early leads to lethal VM phenotypes [124,195,202,203,204,205,206]. To overcome this, researchers have used special approaches to generate PIK3CA-related vascular phenotypes in mouse models. Using patient-derived xenograft (PDX) mouse models has allowed for the investigation of the pathogenesis of venous malformations (VeMs), as activating PIK3CA mutations in endothelial cells have been identified as the main drivers of VeMs [202,203,205,206]. Several VeMs PDX mouse models have been created by injecting human VeM-derived endothelial cells into athymic nude mice. Within 7–9 days, ectatic vascular channels can be seen in the VeM-PDX mouse models. The PDX mouse models have demonstrated that human VeM-ECs with different activating PIK3CA mutations behave like tumor cells and invade mouse tissue to form lesions with human VeM-ECs [20,98,205,207], similar to VeM phenotypes in humans [20,98,205,207].

## 5. Vascular Phenotypes Shared by VeMs in Cancer Mouse Models

*Timing for Expression of GOF PIK3CA point mutations is a challenge*. Mouse models with GOF PIK3CA point mutations have frequently been used in tumorigenesis studies, using various transgenic techniques [174]. These models have primarily focused on targeting epithelial cells, with both overexpression of the activating mutation H1047R [79] and knock-in of mutations E545K or H1047R [163] resulting in vascular phenotypes. However, properly timing the expression of these mutations in VeM mouse models remains challenging. Mouse models of heterozygous GOF PIK3CA point mutation H1047R in endothelial cells resulted in embryonic death with extraembryonic defects [195]. This phenotype is common in VeM/LM mouse models with activating PIK3CA mutations, and likely explains the absence of inherited germline *PIK3CA* mutations in humans. In order to overcome this issue, inducible mouse models with active PIK3CA mutations were created, which resulted in the mice dying from multiple uncontrolled growth abnormalities and visible subcutaneous vascular abnormalities [99,205,207,208], resembling typical VeM symptoms in humans [209].

*VeM-associated metastatic tumor cells or VeM-Endothelial cells (VeM-ECs) carrying activating PIK3CA mutations are invasive*. Activating mutations in key factors of the PI3K/AKT/mTOR pathway, such as PIK3CA, AKT, or MAP3K3, lead to the activation of PI3K signaling and uncontrolled cell proliferation in many human cancers [76]. In PROS, the timing and nature of postzygotic activation of these factors determine the distribution of highly proliferative cells in the individual, resulting in overgrowth, specific tumors, and subcutaneous vascular abnormalities [194,210]. Cancer is a common abnormality in PROS, and it is possible that metastatic forms of tumor cells with activating mutations in PIK3CA or MAP3K3 can invade tissue/vasculature [211], circulate in the blood, and reside in microvascular lesions, including CCMs [5]. Activating mutations in key factors of the PI3K/AKT/mTOR pathway, such as TIE2, PIK3CA, AKT, or MAP3K3, lead to the activation of PI3K signaling and uncontrolled cell proliferation in many human cancers [76]. In PROS, the timing and nature of postzygotic activation of these factors determine the distribution of highly proliferative cells in the individual, resulting in overgrowth, specific tumors, and subcutaneous vascular abnormalities [194,210]. As noted earlier, cancer is a frequent occurrence in PROS, and it is plausible that tumor cells with activating mutations in PIK3CA or MAP3K3 may infiltrate tissues and blood vessels in metastatic forms, leading to the formation of microvascular lesions such as CCMs [211].

## 6. Sporadic CCMs Caused by Somatic Passenger Mutations of *CCM* Genes

*DVAs are a causative factor for sporadic cases of CCMs.* Developmental venous anomalies (DVAs) are a type of venous malformation (VeMs) located in the brain. It is believed that both DVAs and VeMs have a similar genetic cause [98,99,101,212,213]. The coexistence of CCMs and DVAs has long been recognized and is considered a standard criteria in identifying sCCM cases [214,215,216,217,218]. Studies have found a significant link between typical DVAs and sCCMs but not with fCCMs [217,219,220]. This suggests that DVAs may play a role in the development of sCCMs. Recently, a case report highlighted the presence of long-term distant recurrence of hemorrhagic sCCM associated with a DVA, suggesting that focal inflammatory and stressful conditions generated by DVAs could contribute to CCM lesion development or clinical manifestation [221]. It has been noted that although sporadic CCMs are not hereditary, they may present multiple lesions, similar to familial CCMs, leading to misdiagnosis [27]. However, the distribution of these lesions in sporadic cases is different from that in familial cases. While familial CCMs are evenly distributed throughout the brain, sporadic CCMs are often clustered in the area drained by the collecting vein associated with DVAs. This suggests that sporadic CCMs can be considered as VeM-derived CCMs [5]. 

*Sequential events of VeMs-derived CCM lesions in sporadic CCM cases.* Like tumorigenic events observed in breast cancer, somatic point mutations in angiogenic factors such as *TIE2*, *AKT1*, *MAP3K3*, and *PIK3CA*, involved in PI3K signaling, have been identified as a cause of enhanced RTK/PI3K/AKT/mTOR signaling in ECs and play a key role in the pathogenesis of VeMs. Although genetic screening for the three *CCM (1-3)* genes has not found any genomic mutations in sporadic CCM cases associated with VeMs [27,222], somatic mutations in the *CCM (1-3)* genes have been found in CCM lesions along with GOF mutations of the *PIK3CA* gene [1,2,3]. This suggests that VeM-ECs with enhanced PI3K/AKT/mTOR signaling, similar to their breast cancer counterparts, show high proliferative potential, decreased genomic stability, and increased invasiveness. The VeM-derived CCM model remains the most likely explanation for sCCM and DVAs [5], which are linked by a common genetic mechanism and are characterized by bi-allelic mutations in DNA mismatch repair genes in patients with constitutional mismatch repair deficiency syndrome, even though both conditions have their own independent regulatory signaling cascades [223]. The presence of DVA-led CCM lesions around DVAs supports this hypothesis and suggests that GOF mutations of the *PIK3CA* gene drive the subsequent appearance of passenger mutations, including *CCM* genes, in ECs; which can be mirrored in breast and reproductive cancers (Figure 1 and Figure 2). A common genetic paradigm linking reproductive cancers and PROS associated vascular anomalies with the same GOF *PIK3CA* gene mutation should be established [224], while mutated *CCM* genes result in clinically observed sCCM cases.

## 7. Materials and Methods

In our meta-analysis, we used Oncogrid to visualize the genomic landscape of breast and reproductive cancer patients from the Cancer Genome Atlas (TCGA) PanCancer Atlas. The Oncogrid analysis integrated genomic variants and tumor samples, available through the TCGA portal web application [225,226]. For breast cancer, Oncogrid was generated for the top two somatically mutated genes (*TP53* and *PIK3CA*), as well as another frequently mutated gene in the PI3K signaling pathway (*AKT1),* using 200 tumors from a multi-centric cohort. The same process was repeated for reproductive cancer, with Oncogrid generated from 200 tumors and the same three oncogenes. In this paper, our focus was on gain of function (GOF) point mutations in the *PIK3CA* gene, so we only retained simple somatic mutations (SSM) for Oncogrid by excluding copy number variations and alteration-based survival data. The genes were ordered based on their alteration frequency in each tumor, and SSMs were represented as green-colored dots.

## 8. Conclusions

In this meta-analysis, we aimed to examine the coexistence of gain-of-function (GOF) mutations in the *PIK3CA* gene and loss-of-function (LOF) mutations in *CCM* genes in sCCM lesions. We utilized genomic data from the Cancer Genome Atlas (TCGA) for breast and reproductive cancers to inform our analysis. Our goal was to understand the relationship between mutational events and CCM lesions over time. We will perform a comparative meta-analysis for GOF PIK3CA point mutations in an attempt to demonstrate the genetic similarities shared by both cancers and vascular anomalies. Given the well-known role of GOF PIK3CA mutations as a driver oncogene in reproductive cancers, particularly breast cancer, we conducted a comparative meta-analysis to highlight any genetic similarities shared between these cancers and vascular anomalies.

## Figures and Tables

**Figure 1 jpm-13-00673-f001:**
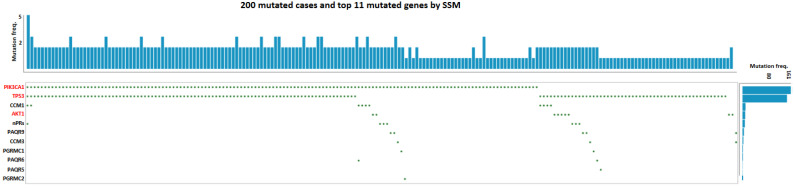
Oncogrid analysis of simple somatic mutations in known breast cancer oncogenes and key genes within the CmPn signaling network. Known driver oncogenes (red colored) and key components (black colored) of the CmPn signaling network from 200 breast cancer lesions in Genomic Data Commons (GDC) and The Cancer Genome Atlas (*TCGA*). Utilizing the GDC data portal from the National Cancer Institute (NCI), the distribution of simple somatic mutations (SSM) for breast cancer lesions for common breast cancer driver genes, along with SSM counted in all key CmPn players were assessed. SSM, represented with green dots, are somatic mutations that include single base substitutions, small deletions, and insertions (≤200 bp).

**Figure 2 jpm-13-00673-f002:**
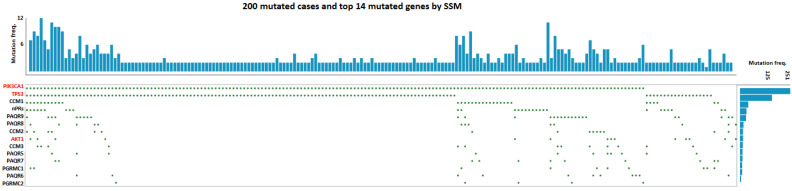
Oncogrid analysis of simple somatic mutations in known reproductive cancer oncogenes and key genes within the CmPn signaling network. Known driver oncogenes (red colored) and key components (black colored) of the CmPn signaling network from 200 breast cancer lesions in Genomic Data Commons (GDC) and The Cancer Genome Atlas (*TCGA*). Utilizing the GDC data portal from the National Cancer Institute (NCI), the distribution of simple somatic mutations (SSM) for reproductive cancer lesions for common cancer driver genes, along with SSM counted in all key CmPn players were assessed. SSM, represented with green dots, are somatic mutations that include single base substitutions, small deletions, and insertions (≤200 bp).

## Data Availability

Data were obtained from the Cancer Genome Atlas (TCGA) and are available from the corresponding author if necessary.

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
