# Peer review of "Familial CCM Genes Might Not Be Main Drivers for Pathogenesis of Sporadic CCMs-Genetic Similarity between Cancers and Vascular Malformations"

_jpm, 2023, doi:10.3390/jpm13040673_

Round 1

Reviewer 1 Report

In this review article, the authors overviewed the similarity between breast and reproductive cancers and sporadic cerebral cavernous malformations (sCCMs) from the viewpoint that both lesions have gain-of-function (GOF) PIK3CA mutations. As a result of consideration, they concluded that sCCMs and those cancers share the genetic features that they have PIK3CA mutations frequently, which shows those mutations are drivers of the lesions, and have mutations in CCM genes less frequently, which suggests those mutations are passenger mutations.

This article described that venous malformations (VeMs) probably arise from GOF PIK3CA mutations, which can prompt the generation of passenger mutations in several genes, including CCM genes, by comparing the genetic landscape of breast and reproductive cancers. Although loss-of-function (LOF) mutations in CCM genes have been identified in sCCMs, the frequency is known to be low. Recently, GOF PIK3CA mutations were identified in sCCM in addition to LOF mutations in CCM genes. However, the role of GOF PIK3CA mutations remains to be understood. So, the consideration of this study is valuable.  

However, there are some issues the authors should address to improve the quality of the manuscript.

1. In line 59, the words vascular malformation is mistakenly abbreviated to VeM. The authors seem to should use VM here.

2. In line 62, the description of the references is not appropriate. The same error is also seen in line 204.

3. The meaning of the heading "GOF PIK3CA mutations prompt the appearance of numerous passenger mutations" in line 143 is almost the same as the former heading "GOF PIK3CA point mutations can generate many passenger mutations in breast cancers" in line 130. The authors should change either or both of the headings. In the first place, the text following the heading "GOF PIK3CA mutations prompt the appearance of numerous passenger mutations" does not describe passenger mutations at all. The authors must reconfigure these two paragraphs' headings and/or descriptions.

4. The description from line 153 to line 159 seems improper. I need help understanding the reason why the authors describe rapamycin here. Maybe, so do the readers. I think it is reasonable that some descriptions of the CCM signaling complex or CmPn signaling network are located here, in the first paragraph following the heading "GOF PIK3CA mutations induce the passenger mutations of members of CmPn signaling in reproductive cancers. I recommend some reconsideration of the composition here.

5. The sentence "Cancer is a common abnormality in PROS and it is possible that metastatic forms of tumor cells with activating mutations in PIK3CA or MAP3K3 can invade tissue/vasculature, circulate in the blood, and reside in microvascular lesions, including CCMs" appears twice in the lines 266 and 275.

6. The way of using abbreviations, such as fCCM and sCCM, are inappropriate. The abbreviations should be defined at first mention and used consistently thereafter.

7. The names of genes should be written in italics.

8. At least the figure titles and possibly figure legends have to be given to the figures. 

Reviewer 2 Report

Comments to the Manuscript “jpm-2325290” entitled “Familial CCM genes might not be main drivers for pathogenesis of sporadic CCMs-genetic similarity between cancers and vascular malformations” by Jun Zhang et al.

The manuscript by Jun Zhang et al. describes a meta-analysis that aims to analize the coexistence of gain-of-function (GOF) mutations in PIK3CA and loss-of-function (LOF) mutations in CCM genes in the sporadic form of the disease (known as sCCM). The authors utilized genomic data from the Cancer Genome Atlas (TCGA) for different cancer types in order to understand the relationship between mutational events and CCM lesions over time, and demonstrate that GOF point mutations of PIK3CA show significant similarities shared by both cancers and vascular malformations (here indicated as VeMs).

 General comments:

The manuscript is overall well written and sounds scientifically fine. This review highlights some fundamental aspects of point mutations shared by both cancers and cerebrovascular disease. Indeed, it points out that gain-of-function mutation in specific genes may induce the proliferation of endothelial cells, as well leads to genomic instability and an increased number of secondary mutations, as it happens in the cancer cells during the evolution of the cancer. Therefore, these mutations are drivers for vascular anomalies and contribute to clinical symptoms, and they also play a key role in recurrent and simultaneous manifestations of CCCM lesions and different VeMs. As minor comment, some connections could be better showed and described in order to make more clear the network established between the described mutations and the affected signaling pathways, and CCM or cancer progression. Also, a short schematic or summarizing figure might be helpful to increase the quality of the paper and its topic.

Specific comments:

1)      In the Introduction section, as well as in Sub-title “Sporadic CCMs caused by somatic passenger mutations of CCM genes”, additional sentence should be added demonstrating the tight relationship between CCM and DVA, as well a DVA can induce recurrence in hemorrhagic events from a sCCM, “Recently, a case report highlighted the presence of long-term distant recurrence of hemorrhagic sCCM associated with a DVA, suggesting that focal inflammatory and stressful conditions generated by DVAs could contribute to CCM lesion development or clinical manifestation (PMID: 36498972).

2)      Line 62: few references have not been added properly.

3)      Figure 1 and Figure 2 are both missing of a caption, and some genes are difficult to read.

4)      Sub-title “The oncogenic driver role of GOF PIK3CA mutations was confirmed in cancer mouse models.” could be improved in style.

5)      Lines 204-205: few references have not been added properly.

6)      Line 230: Pik3ca should be written in capital letters.

7)      Vascular malformations are abbreviated both as VeMs and VM: please use one abbreviation only.

8)      Line 272: reference (Lai et al., 2015) has not been added properly.

9)      Line 274: few references have not been added properly.

10)  Lines 275-278 are repetition of lines 266-269. It could be better to remove or change them to make the focus easier.
